# Predicting One-Year Deaths and Major Adverse Vascular Events with the Controlling Nutritional Status Score in Elderly Patients with Non–ST-Elevated Myocardial Infarction Undergoing Percutaneous Coronary Intervention

**DOI:** 10.3390/jcm10112247

**Published:** 2021-05-22

**Authors:** Muhsin Kalyoncuoğlu, Fahrettin Katkat, Halil Ibrahim Biter, Sinem Cakal, Aydin Rodi Tosu, Mehmet Mustafa Can

**Affiliations:** 1Cardiology Department, Haseki Training and Research Hospital, University of Health Sciences Turkey, Istanbul 34096, Turkey; abrahambiter@hotmail.com (H.I.B.); sinemdnz@gmail.com (S.C.); aydinroditosu@gmail.com (A.R.T.); mehmetmustafacan@yahoo.com (M.M.C.); 2Cardiology Department, Bagcilar Training and Research Hospital, University of Health Sciences Turkey, Istanbul 34200, Turkey; fahrettin_katkat@hotmail.com

**Keywords:** aged, myocardial infarction, malnutrition, mortality, morbidity

## Abstract

The prognostic value of malnutrition in elderly patients with non-ST-elevated myocardial infarction (NSTEMI) is not fully understood. Nutritional characteristics were evaluated by novel Controlling Nutritional status (CONUT), the prognostic nutritional index (PNI) and the geriatric nutritional risk index (GNRI) scores. The impact of these scores on major outcomes in 253 NSTEMI patients over 60 years and older were assessed. Compared to those with good nutritional status; malnourished patients had more major adverse cardiac and cerebrovascular events (MACCEs) at 1-year follow up. Multivariable cox regression analysis revealed that CONUT (hazard ratio = 1.372; *p* < 0.01) was independent predictor of MACCEs, whereas PNI (*p* = 0.44) and GNRI (*p* = 0.52) were not. The discriminating power of the CONUT (AUC: 0.79) was adequate and significantly superior to both the PNI (AUC: 0.68) and the GNRI (AUC: 0.60), with a *p*-value for both < 0.01. Patients with elevated CONUT exhibited the highest event rate for all-cause mortality and MACCEs in survival analysis (*p* < 0.01). We conclude that malnutrition is strongly associated with adverse outcomes in older patients with NSTEMI. In fact, the CONUT score adequately predicts one-year MACCEs among elderly NSTEMI patients who achieve complete revascularization after coronary intervention.

## 1. Introduction

The mortality and morbidity rates among patients with acute coronary syndrome (ACS) have decreased significantly due to medical breakthroughs, but advances in ACS management have not equally improved outcomes between younger and older individuals. Moreover, aged patients with non-ST-elevated ACS (NSTE-ACS) less frequently undergo invasive procedures in comparison with younger ones due to the concerns of both patients and physicians about the increased risk of complications [1,2]. Therefore, when deciding on the management course for NSTE-ACS in these elderly patients, in addition to considering the estimated risks and benefits of revascularization therapy, the patient’s life expectancy and any accompanying comorbidities should also be taken into account [3]. As such, the identification of high-risk patients by modifiable clinical features may be helpful for physicians to improve prognosis and clinical outcomes in elderly patients.

Malnutrition is an important and modifiable clinical parameter that adversely affects the health of older individuals but can be treated by physicians [4]. Recently, it has been reported that malnutrition is associated with the development of atherosclerosis and a greater incidence of cardiovascular mortality in elderly patients [5]. It has also been shown to be an important prognostic factor for several cardiovascular diseases (CVDs), such as heart failure, hypertension, valvular heart disease and atrial fibrillation [6,7,8,9,10]. Although malnutrition has also been studied in patients with ACS, these studies largely included either ST-elevated myocardial infarction (MI) (STEMI) patients who underwent primary percutaneous coronary intervention (PCI) or heterogeneous ACS cohorts [11,12]. To our knowledge, no specific data exist to suggest the prognostic accuracy of malnutrition in elderly non-STEMI (NSTEMI) patients who achieved complete revascularization (CR) with PCI. Therefore, in this study, we sought to investigate the predictive role of malnutrition-based scoring systems, including the Controlling Nutritional Status Score (CONUT), the Prognostic Nutritional Index (PNI) and the Geriatric Nutritional Risk Index (GNRI), in determining one-year outcomes in this specific population.

## 2. Materials ant Methods

### 2.1. Study Population

The study flowchart and exclusion criteria are summarized in Figure 1. The diagnosis of ACS was made according to current clinical practice guidelines [13].

Since the United Nations criterion for old age is 60 years and our country is one of the founding members of the United Nations, we used this cutoff value, keeping in line with the actions of some previous research [14]. Patient baseline clinical and demographic characteristics and laboratory parameters, including total cholesterol, serum albumin and lymphocyte count, were obtained. All patients were treated in accordance with current guidelines [13]. After hospital discharge, follow-up was conducted by direct contact or telephone interview with the patient or, if the patient was deceased, by discussion with family members. The national death notification system and hospital records were also used to obtain information on mortality. Since our study was retrospectively designed, written informed consent from the participants could not be obtained, but our study protocol conformed to the principles of the Declaration of Helsinki and was approved by the local ethics committee of our institution.

### 2.2. Angiographic Analysis

Coronary angiograms were recorded to digital media for quantitative analysis (DICOM viewer; MedCom GmbH, Darmstadt, Germany). Coronary angiograms were analyzed by two experienced interventional cardiologists blinded to the study participants’ clinical and laboratory data. CAD was defined as a finding of stenosis of more than 50% of the lumen diameter in any of the main coronary arteries. From the baseline diagnostic angiogram, the anatomic and clinical severity of coronary stenosis were quantitatively evaluated using the Synergy Between Percutaneous Coronary Intervention with TAXUS and Cardiac Surgery (SYNTAX) score I (SS I) and II for PCI (SSII-PCI) by using the downloadable version hosted at http://www.syntaxscore.org (accessed on 1 April 2021).

Success of PCI was determined by achieving the anatomical complete revascularization which is defined as successful treatments of all coronary artery lesions or segments > 1.5 mm in diameter with ≥50% narrowing of the lumen, irrespective of their functional significance, during the period of index hospitalization [15].

### 2.3. Nutritional Status Measurement Tools

Body mass index, defined as the body weight (in kilograms) divided by the square of height (in meters), was calculated for all patients.

The CONUT score was calculated using the serum albumin, total cholesterol and total lymphocyte count. Details of this scoring system are summarized in Table 1 [15]. The study cohort was divided into two groups of those with and without malnutrition, respectively. Patients scoring zero to one point(s) were classified into the normal (non-malnourished) group, while those who scored two points or more were considered to be malnourished [16].

The PNI score was calculated using the following formula: 10 × serum albumin value (g/dL) + 0.005 × total lymphocyte count (per mm3). A score of greater than 38 points was defined as normal [17].

The GNRI was developed by modifying the nutritional risk index for elderly patients and scores were calculated using the following equation: (1.489 × serum albumin (g/dL)) + 41.7 × (present weight (kg)/ideal body weight (kg) [18]. The ideal body weight was calculated from the Lorentz equations and, when greater than or equal to 1, the calculation was performed by setting the ratio to 1 [7,12]. Patients with GNRI scores of 98 points or above were considered as low risk [7].

### 2.4. Study Endpoint

The prespecified endpoint of this trial was major adverse cardiac and cerebrovascular events (MACCEs), defined as a composite of all-cause death, any myocardial infarction, any revascularization and any stroke, within one year of the follow-up period according to the Academic Research Consortium-2 consensus [19].

### 2.5. Statistical Analysis

Continuous variables were presented as mean ± standard deviation values if normally distributed and median (interquartile range (IQR)) values if not normally distributed, while categorical variables were given as percentages. The chi-squared (χ^2^) test was used to compare categorical variables between the groups, and the Kolmogorov–Smirnov test was employed to assess whether the variables were normally distributed. A Student’s t-test or Mann–Whitney U test was used to compare the continuous variables between groups according to whether they were normally distributed or not. Pearson’s correlation coefficient was calculated to describe the degree of correlation between the malnutrition-based scores and CAD severity. To determine the independent predictors of one-year MACCEs, variables found to be associated at a *p* < 0.05 level according to univariate analysis were included in the multivariate Cox regression analysis, with the results reported as the hazard ratios (HR) and 95% confidence intervals (CIs). The capacity to discriminate between patients with and without MACCEs was determined using the receiving operating characteristic (ROC) curve and area under the ROC curve (AUC), accompanied by 95% CIs. Discriminatory power was classified as ‘good’ if the AUC was 0.70 or greater and as inadequate if the AUC was less than 0.70 [20]. To compare the predictive performance of the aforementioned scores, the pairwise comparison of ROC curves was performed using the method of DeLong et al. [21]. The optimal cutoff value was calculated from the points of maximal sensitivity and specificity by using Youden’s index [22]. Time-to-event data were presented graphically by using Kaplan–Meier survival curves and log-rank tests. The threshold of statistical significance was established at *p* < 0.05. All statistical analyses were performed using the Statistical Package for the Social Sciences version 24.0 software program (IBM Corp., Armonk, NY, USA). ROC curves of the models were compared using the MEDCALC software program (MedCalc Software bv, Ostend, Belgium).

## 3. Results

### 3.1. Clinical and Laboratory Characteristics of Malnourished Patients

Among 253 study participants, malnutrition was present in 6.3% (*n* = 16 patients) with PNI, 38.7% (*n* = 98 patients) with CONUT scores and 64% (*n* = 160 patients) with GNRI scores. According to their CONUT scores, 90 (36%) patients had mild malnutrition and eight (3.2%) had moderate to severe malnutrition. Malnourished patients tended to be older (70.9 ± 7.3 vs. 66.9 ± 6.1 years; *p* < 0.01) with statistically lower body mass index values (27.2 ± 3.0 vs. 28.5 ± 2.6 kg/m^2^; *p* < 0.01), more frequent chronic heart failure (22.4% vs. 11%; *p* = 0.02) and lower left ventricular ejection fraction values (46.5 ± 8.4 vs. 50.9 ± 6.4; *p* < 0.01) than those without malnutrition. Although it was only statistically borderline significant, a history of CAD was also more common among malnourished patients (36.7% vs. 26.5%; *p* = 0.08). Furthermore, malnourished patients had higher Killip class (24.5% vs. 6.5%; *p* < 0.01) and Global Registry of Acute Coronary Events risk scores (127 ± 21.4 vs. 113.6 ± 14.1 points; *p* < 0.01) than non-malnourished patients. Considering laboratory examination results, individuals suffering from malnutrition had lower GFR values (74 ± 21 vs. 82 ± 18; *p* < 0.01), lower serum total cholesterol levels (192 ± 44 vs. 217 ± 38; *p* < 0.01), lower albumin levels (35.4 ± 3.1 vs. 38.7 ± 3.1; *p* < 0.01) and lower lymphocyte counts (1.2 (0.9–1.8) vs. 2.1 (1.7–2.5); *p* < 0.01) relative to participants who were not malnourished. When both groups were evaluated using the other two scoring systems based on malnutrition, malnourished patients had statistically lower PNI (42.8 ± 3.8 vs. 49.7 ± 5.4 points; *p* < 0.01) and GNRI (94.7 ± 4.6 vs. 99.3 ± 4.6 points; *p* < 0.01) scores. Malnourished individuals also had higher anatomical SSI (15 (11–24) vs. 10.0 (7–15) points; *p* < 0.01) and higher SSII-PCI (34 (28–43) vs. 27 (23–33) points; *p* < 0.01) points than the non-malnourished patients. Detailed demographic, clinical parameters and laboratory and angiographic parameters of all study participants and as compared between the two groups are included in Table 2 and Table 3.

Additionally, correlation analysis for the relationship between malnutrition scores and CAD severity revealed that the CONUT score exhibited a moderately positive correlation (r = 0.352; *p* < 0.01), while PNI (r = −0.190; *p* < 0.01) and GNRI (r = −0.167; *p* < 0.01) scores were negatively and weakly correlated with the anatomical SSI, which is a quantitative indicator of CAD severity. On the other hand, CONUT (r = 0.438; *p* < 0.01), PNI (r = −0.333; *p* < 0.01) and GNRI (r = −0.316; *p* < 0.01) scores showed moderate correlations with the SSII-PCI score. In addition, CONUT (r = 0.445; *p* < 0.01), PNI (r = −0.316; *p* < 0.01) and GNRI (r = −0.261; *p* < 0.01) scores were moderately correlated with the Global Registry of Acute Coronary Events risk score predicting six-month mortality for patients with ACS.

### 3.2. Factors Associated with One-Year Major Adverse Cardiac and Cerebrovascular Events

During the follow-up period (mean length of 20.5 ± 9.2 months), among malnourished individuals, one-year mortality occurred in 22 (22.4%) patients, and one-year MACCEs were observed in 36 (36.7%) patients. Of the deceased patients, 2 had cardiac arrest of unknown cause, 2 had documented fatal ventricular arrhythmia, 5 had decompensated heart failure or acute pulmonary edema, and 13 had myocardial infarction. In addition, among patients who had MACCEs, except for those who died during the one-year follow-up period, recurrent revascularization was performed in four patients due to stable angina, two due to unstable angina and eight due to non-fatal MI. Furthermore, of these, two patients had stent thrombosis, seven had in-stent restenosis and 17 underwent revascularization for the vessels other than the infarct artery. No patient experienced stroke. Additionally, 14 (14.3%) of those with malnutrition died within 30 days after the index NSTEMI. As 30-day mortality was observed in a relatively small number of patients, we did not conduct a statistical analysis for 30-day outcomes.

To determine the independent predictors of one-year MACCEs, we performed a multivariate cox regression analysis by including variables that were significantly associated with MACCEs in the univariate analysis. As total cholesterol level, serum albumin level and lymphocyte count were already considered by the malnutrition-based scores, these variables were not taken into account in the multivariate analysis that included CONUT and PNI scores, regardless of their significance in the univariate analysis. Instead, total cholesterol level, lymphocyte count and serum albumin level were evaluated in a separate multivariate analysis model (model 1) that did not include the aforementioned scores (Table 4). GNRI was also not included in the multivariate analysis as it was not statistically associated with MACCEs in the univariate analysis (*p* = 0.09). Moreover, to avoid overfitting the model, CONUT and PNI scores were not included in the same multivariate Cox regression analysis model and were evaluated in separate models (models 2 and 3, respectively) (Table 5).

In the model 1 analysis, lymphocyte count was found to be an independent predictor for MACCEs, but total cholesterol and albumin levels were not. In all multivariate Cox regression analysis models, the presence of diabetes and low left ventricular ejection fraction were found to be independent predictors of one-year MACCEs. While the CONUT score independently predicted MACCEs in model 2 (HR = 1.434; *p* < 0.01), the PNI score did not do so in model 3, with a *p*-value of 0.43 (Table 5).

A comparative analysis of ROC curves revealed that the discriminating ability of the CONUT score (AUC: 0.79, 95% CI: 0.73–0.84; *p* < 0.01) was adequate and significantly superior to both that of the PNI score (AUC: 0.68, 95% CI: 0.62–0.74; *p* < 0.01) and the GNRI score (AUC: 0.60, 95% CI: 0.54–0.67; *p* = 0.04), with a *p*-value for both < 0.01 (Figure 2). Additionally, the discriminatory power of the PNI score was better than that of the GNRI score (*p* = 0.04). A CONUT score cutoff value greater than 2 points (56% sensitivity and 92% specificity), a PNI score cutoff value of 41.2 points or less (42% sensitivity and 91% specificity), and a GNRI score cutoff value of 92.3 points or less (42% sensitivity and 89% specificity) predicted the one-year MACCEs. The Kaplan–Meier curves in Figure 3 represent the one-year adverse outcomes in patients stratified into low-risk and high-risk groups based on the determined cutoff values.

## 4. Discussion

The prevalence of malnutrition increases with age and has been reported at rates varying from 29% to 61% in different series [23]. Similar to the results of the study by Roubin et al., in our study of patients with ACS, between 40% and 60% of participants were classified as malnourished based on CONUT and GNRI scores, while only 6.3% of them had malnutrition according to the PNI score distribution [12]. In elderly patients with CAD, malnutrition is common due to the effects of concomitant disease, stress reaction, drug usage, anxiety or depression and is independently associated with the risk of all-cause death [23]. Accordingly, elderly patients diagnosed with ACS should be screened for the presence of malnutrition, and its treatment and prevention are important challenges for health care providers to overcome.

Several scoring systems, such as the CONUT score, PNI score and GNRI score, have been proposed as markers to reflect the malnutrition status [11,12,24]. Although no nutritional index has yet been firmly established in patients with CAD, few reports are available on the prognostic significance of the aforementioned scores in patients with ACS [11,12,25,26,27]. Additionally, according to some studies that evaluated the relationship between the CONUT score and prognosis in several CVDs, the prognostic value of the CONUT score in patients with ACS has rarely been discussed in the literature [6,7,8,9,11,12,24]. In their study, Basta et al. evaluated the prognostic impact of CONUT and PNI scores in elderly STEMI patients undergoing primary PCI [11]. As seen in our study, they found that the CONUT score was associated with an increased risk of all-cause death for both the unadjusted model and age- and sex-adjusted model, but the PNI score was not. However, it is necessary to consider that the ability of the CONUT score to predict mortality disappeared after adjusting for all mortality-related parameters in the multivariate analysis. It should be taken into account that the percentage of patients with malnutrition in this study was quite high (>80%) relative to the literature data, and the results of this study should be interpreted carefully. In another study, Roubin et al. evaluated the prognostic significance of the CONUT, PNI and GNRI scores in patients with ACS [12]. Similar to the results of our study, they observed that the CONUT score was associated with a poor prognosis after adjustment for all MACCE-related parameters (e.g., age, type of ACS, PCI and complete revascularization). Meanwhile, although both GNRI and PNI were associated with MACCEs in their study, they reported that the prediction ability of these two indices was lower than that of the CONUT score. In addition, PNI showed a significant prognostic value in STEMI patients undergoing primary PCI in a study conducted by Chen et al., but these authors suggested that PNI may not be reliable for predicting mortality in patients with acute MI, similar to our findings [26,27]. In terms of the predictive power of the CONUT score relative to that of the other two scores, the results of the present study support the findings of the study conducted by Roubin et al. Although these authors adjusted for other risk factors, their study included a large and heterogeneous ACS population, including younger patients and individuals with unstable angina, STEMI or with incomplete revascularization. However, malnutrition is a complex issue due to diversity in etiology and a wide range of determinants, especially in older adults. Additionally, there are data that a high proportion of older adults are at risk for malnutrition, with estimates ranging from 20% to 50%, although prevalence estimates vary substantially depending on the population considered [4]. We also know that, as compared with STEMI patients, NSTEMI patients have a more complex clinical phenotype, including older age and more comorbid pathologies, and, as a result, long-term outcomes in NSTEMI patients do not typically improve to the extent seen in STEMI patients. This complexity affects clinical decision-making, especially in high-risk NSTEMI patients for whom risk–benefit assessments are problematic. Therefore, newly defined modifiable clinical features, such as the CONUT score, may be helpful in decision-making and improving clinical outcomes, especially in older NSTEMI patients. As far as we know, no studies have specifically addressed this issue in elderly patients with NSTEMI who achieved successful reperfusion with PCI, and the present study appears to be the first to investigate the predictive value of the CONUT score in determining one-year MACCEs in this specific group.

On the other hand, unlike in previous studies, we did not observe the predictive ability of both GNRI and PNI in our study. The GNRI score is calculated using the serum albumin level, measured body weight and ideal body weight [17]. In terms of the GNRI score, our contradictory finding, which is inconsistent with those of previous studies, may be related to the change in body weight affected by fluid status, leading to an erroneous prediction of nutritional status in elderly patients with NSTEMI. The CONUT score includes the serum albumin level, total cholesterol level and total lymphocyte count for the assessment of nutritional status, while the PNI index only includes serum albumin level and lymphocyte count. The albumin level, total cholesterol level and lymphocyte count were significantly lower in patients with MACCEs than in those without MACCEs and were associated with poorer outcomes. Furthermore, in line with the literature, lymphopenia was found to be an independent predictor of adverse cardiovascular outcomes in our study. Indeed, there is also substantial evidence to suggest that decreased albumin plasma concentrations and a relatively low lymphocyte count have prognostic value in MI [28,29]. Therefore, we think the fact that the CONUT score includes all the parameters mentioned above may explain why it has a better predictive value than the other two scoring systems considered.

One of the most interesting points of our study is the existence of an inverse relationship between total cholesterol level, which is a parameter of the CONUT score, and mortality. Although the findings of this study contradict the general assumption that hypercholesterolemia is associated with adverse cardiovascular outcomes, previous studies investigating the prognostic value of plasma lipids at admission observed paradoxically better outcomes in hypercholesteremic patients with ACS, referred to as the ‘cholesterol paradox’ [30,31]. As people age, the concurrent increase in non-CVDs, which lowers cholesterol levels and increases the risk of death, may be one of the reasons for this inverse relationship [32]. Moreover, shadowing of the predictive effect of hypercholesterolemia in the presence of stronger risk factors may be another reason as alluded to in previous studies where similar findings were discussed [33].

The mechanisms underlying this relationship between malnutrition and poor prognosis in patients with CVDs have not yet been established, although inflammation may be one possible mechanism. Inflammation has been proposed to be associated with the development and progression of atherosclerosis and was found to be correlated with unfavorable outcomes among patients with CVD [28,29]. It is also associated with anorexia and the catabolism of skeletal muscle and adipose tissue, which may contribute to the nutritional compromise, muscle weakness and weight loss that characterize frailty [34]. The relationship between these three entities has recently been labelled as malnutrition inflammation atherosclerosis syndrome [35].

Malnutrition is a complex condition characterized by reduced protein reserves, calorie breakdown and weakened immune defenses. The PNI and CONUT scores, unlike the GNRI score, include lymphocyte count, which reflects the immune function of the body [8]. Therefore, malnutrition also emerges as a common cause of secondary immunological dysfunction [12]. Failure to recognize that malnutrition can lead to increased morbidity and mortality by causing nutrition-related immunodeficiency and susceptibility to infection is a concern. This point may be considered as another reason for the causality between malnutrition and poor outcome. It may also explain why CONUT and PNI scores have a better predictive value than GNRI.

## 5. Limitations

The present study has some limitations. First, it employed a retrospective design, included a relatively small sample size and relied on single-center experience. Moreover, we could not classify the malnourished patients as mild, moderate or severe due to the small volume of our study. Second, we did not compare the prognostic value of all three nutritional screening tools with more complex comprehensive nutritional assessment tools, such as Subjective Global Assessment and Mini-nutritional Assessment. Third, a nutritional assessment was conducted only at admission, and we did not investigate changes in nutritional status at multiple time points nor their relationship with cardiovascular outcomes. Fourth, the presence of confounding clinical conditions that may be associated with the development of malnutrition, such as undetected cancer, psychiatric disorders and hypothyroidism, was not investigated. Finally, only complete anatomical revascularization was evaluated as indicative of treatment success, and we could not perform a functional evaluation due to technical inadequacy.

## 6. Conclusions

The CONUT score is effective and very easy to calculate for the early detection of malnutrition, even without the use of specific automatic calculators or complex formulas, relative to the PNI and GNRI scores. The current study revealed that high CONUT scores at admission are a key predictor of one-year MACCEs among elderly NSTEMI patients who achieved complete revascularization. We think that, in elderly NSTEMI patients, the CONUT score can help health care professionals to assess patients’ nutritional risk and to identify high-risk individuals who may benefit from nutritional support.

## Figures and Tables

**Figure 1 jcm-10-02247-f001:**
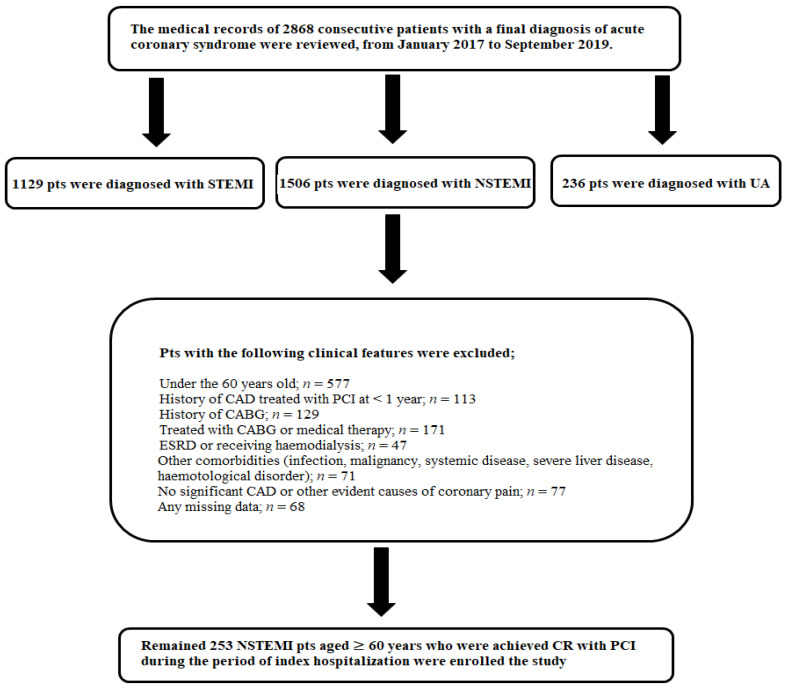
Study flow chart and exclusion criteria. Pts: patients; STEMI: ST-elevated myocardial infarction; NSTEMI: non-ST-elevated myocardial infarction; UA: unstable angina; CAD: coronary artery disease; PCI: percutaneous coronary intervention; CABG: coronary artery bypass grafting; ESRD: end stage renal disease; CR: complete revascularization.

**Figure 2 jcm-10-02247-f002:**
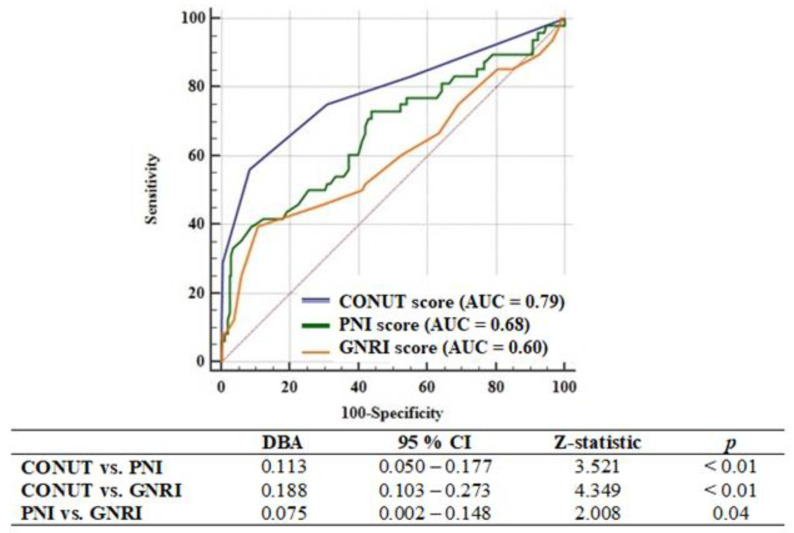
The receiving operating characteristic curves of the CONUT (blue), PNI (green) and GNRI (orange) scores for detecting the one-year MACCEs. CONUT: the Controlling Nutritional Status; PNI: the Prognostic Nutritional Index; GNRI: the Geriatric Nutritional Risk Index; AUC: area under the curve; DBA: difference between areas; CI: confidence interval.

**Figure 3 jcm-10-02247-f003:**
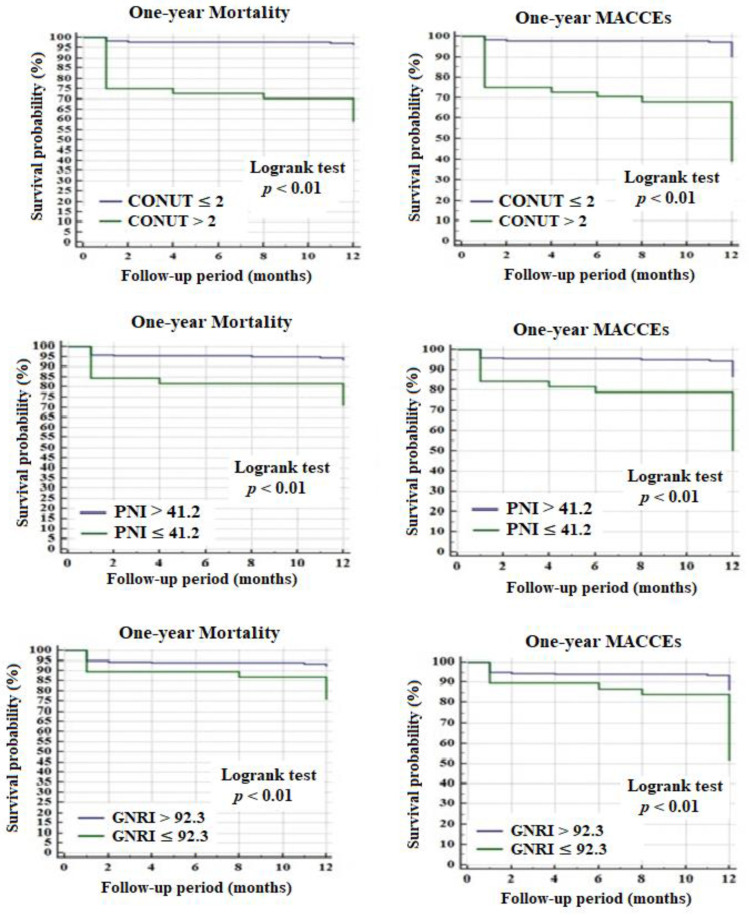
Kaplan–Meier plots of clinical outcomes of the high risk patients categorized by CONUT, PNI and GNRI scores. Blue line means low-risk individuals and green line means high risk individuals. MACCEs: major adverse cardiac and cerebrovascular events; CONUT: the Controlling Nutritional Status; PNI: the Prognostic Nutritional Index; GNRI: the Geriatric Nutritional Risk Index.

**Table 1 jcm-10-02247-t001:** Assessment of malnutrition by CONUT score.

Parameter	Normal	Light	Moderate	Severe
Serum Albumin (g/dL)	3.5–4.5	3.0–3.49	2.5–2.9	<2.5
Score	0	2	4	6
Total Lymphocytes (10^9^/L)	>1.60	1.20–1.59	0.80–1.19	<0.80
Score	0	1	2	3
Total Cholesterol (mg/dL)	>180	140–180	100–139	<100
Score	0	1	2	3
Total Score	0–1	2–4	5−8	9–12

**Table 2 jcm-10-02247-t002:** Demographic and clinical parameters of the study cohort.

Variables	All Population (*n* = 253)	Non-Malnourished (*n* = 155)	Malnourished (*n* = 98)	*p*-Value
Male gender, *n* (%)	181 (71.5)	112 (72.3)	69 (70.4)	0.75
Age, years, ± SD	68.5 ± 6.9	66.9 ± 6.1	70.9 ± 7.3	<0.01
BMI, kg/m^2^, ± SD	28 ± 2.8	28.5 ± 2.6	27.2 ± 3.0	<0.01
Hypertension, *n* (%)	136 (53.8)	80 (51.6)	56 (57.1)	0.39
Diabetes mellitus, *n* (%)	74 (29.2)	41 (26.5)	33 (33.7)	0.22
Dyslipidemia, *n* (%)	156 (61.7)	91 (58.7)	65 (66.3)	0.23
Smoking, *n* (%)	94 (37.2)	62 (40)	32 (32.7)	0.24
Family history, *n* (%)	93 (36.8)	52 (33.5)	41 (41.8)	0.18
CAD history, *n* (%)	77 (37.4)	41 (26.5)	36 (36.7)	0.08
CHF history, *n* (%)	38 (15)	17 (11)	21 (22.4)	0.02
Killip III-IV, *n* (%)	34 (13.4)	10 (6.5)	24 (24.5)	<0.01
LVEF,%, ± SD	49.2 ± 7.6	50.9 ± 6.4	46.5 ± 8.4	<0.01
Grace risk score, ± SD	118.8 ± 18.5	113.6 ± 14.1	127 ± 21.4	<0.01
Syntax Score I, (IQR)	12 (8–18)	10 (7–15)	15 (11–24)	<0.01
Syntax Score II for PCI, (IQR)	29 (24–37)	27 (23–33)	34 (28–43)	<0.01
30-day Mortality, *n* (%)	15 (5.9)	1 (0.6)	14 (14.3)	<0.01
One-year Mortality, *n* (%)	26 (10.3)	4 (2.6)	22 (22.4)	<0.01
One-year MACCEs, *n* (%)	48 (19)	12 (7.7)	36 (36.7)	<0.01
Medications, *n* (%)				
Acetylsalicyclic acid	90 (35.6)	51 (32.9)	39 (39.8)	0.27
ADP receptor antagonists	14 (5.5)	7 (4.5)	7 (7.1)	0.37
Anticoagulant	21 (8.3)	11 (7.1)	10 (10.2)	0.39
Beta-blockers	82 (32.4)	45 (29)	37 (37.8)	0.15
ACEI	67 (26.5)	38 (24.5)	29 (29.6)	0.37
ARB	59 (23.3)	33 (21.3)	26 (26.5)	0.34
CCBs	54 (21.3)	31 (20)	23 (23.5)	0.51
Anti-anginal agents	24 (9.5)	11(7.1)	13 (13.3)	0.10
Statin	57 (22.5)	31 (20)	25 (25.5)	0.30
Fibrats	26 (10.3)	17 (11)	9 (9.2)	0.65
OADs	71 (28.1)	40 (25.8)	31 (31.6)	0.32
Insulin	27 (10.7)	14 (9)	13 (13.3)	0.29

Abbreviations: BMI, body mass index, CAD, coronary artery disease; CHF, chronic heart failure; LVEF, left ventricular ejection fraction; PCI, percutaneous coronary intervention; MACCEs, major adverse cardiac and cerebrovascular events; ADP, adenosine diphosphate; ACEI, angiotensin converting enzyme inhibitor; ARB, angiotensin receptor blocker; CCB, calcium channel blocker; OAD, oral antidiabetic agent.

**Table 3 jcm-10-02247-t003:** Laboratory parameters, and nutrition based scores of the study population.

Variables	All Population (*n* = 253)	Non-Malnourished (*n* = 155)	Malnourished (*n* = 98)	*p*-Value
FBG, mg/dL, (IQR)	123 (102–169)	119 (102–166)	127 (103–178)	0.29
eGFR, mL/min/1.73 m^2^, ±SD	79 ± 20	82 ± 18	74 ± 21	<0.01
Total cholesterol, mg/dL, ±SD	207 ± 42	217 ± 38	192 ± 44	<0.01
LDL-C, mg/dL, ±SD	135 ± 35	142 ± 33	124 ± 35	<0.01
HDL-C, mg/dL, ±SD	43 ± 10	44 ± 10	41 ± 11	0.02
Triglyceride, mg/dL, (IQR)	145 (104–195)	149 (112–208)	135 (98–183)	<0.01
Albumin, g/L, ±SD	37.4 ± 3.5	38.7 ± 3.0	35.3 ± 3.2	<0.01
Haemoglobin, g/dL, ±SD	13.0 ± 1.9	13.6 ± 1.6	12.6 ± 2.2	<0.01
Neutrophil, 10^3^/μL, (IQR)	6.0 (4.4–8.1)	5.7 (4.3–7.9)	6.2 (4.7–8.6)	0.01
Lymphocyte, 10^9^/L, (IQR)	1.9 (1.3–2.4)	2.1 (1.7–2.5)	1.2 (1.0–1.8)	<0.01
Platelet, 10^9^/L, ±SD	234 ± 71	235 ± 74	233 ± 66	0.81
CRP, mg/dL, (IQR)	6.9 (4.0–13)	6.4 (3.8–11.9)	9.3 (4.1–18.1)	0.55
PNI score, ±SD	46.9 ± 5.9	49.6 ± 4.5	42.7 ± 4.0	<0.01
GNRI score, ±SD	97.5 ± 5.2	99.4 ± 4.5	94.5 ± 4.8	<0.01

Abbreviations: FBG, fasting blood glucose; eGFR, estimated glomerular filtration rate; TC, total cholesterol, LDL-C, low-density lipoprotein cholesterol, HDL-C, high-density lipoprotein cholesterol; CONUT, The Controlling Nutritional Status, PNI, Prognostic Nutritional Index, GNRI, Geriatric Nutritional Risk Index.

**Table 4 jcm-10-02247-t004:** Unadjusted univariable and age-adjusted multivariable cox proportional risk regression analysis (without malnutrition-based scores) for determining the predictors of the one year MACCEs.

	Univariate		Model 1 Multivariate	
Variables	HR (95%CI)	*p*-Value	HR (95%CI)	*p*-Value
Age	1.055 (1.015–1.097)	<0.01	1.028 (0.986–1.072)	0.19
BMI	0.886 (0.798–0.985)	0.03	1.009 (0.899–1.134)	0.88
Diabetes mellitus	2.172 (1.231–3.834)	<0.01	1.995 (1.115–3.570)	0.02
LVEF	0.888 (0.857–0.919)	<0.01	0.890 (0.851–0.931)	<0.01
eGFR	0.979 (0.966–0.992)	<0.01	0.996 (0.980–1.012)	0.61
Total cholesterol	0.992 (0.985–0.999)	<0.01	0.998 (0.991–1.004)	0.46
Lymphocyte	0.473 (0.301–0.744)	0.03	0.624 (0.402–0.968)	0.04
Albumin	0.907 (0.830–0.990)	<0.01	1.066 (0.966–1.177)	0.20
CONUT Score	1.731 (1.503–1.993)	0.03	-	-
PNI score	0.918 (0.868–0.970)	<0.01	-	-
GNRI score	0.951 (0.896–1.009)	0.09	-	-

Abbreviations: MACCEs, major adverse cardiac and cerebrovascular events; BMI, body mass index; LVEF, left ventricular ejection fraction; eGFR, estimated glomerular filtration rate; CONUT, The Controlling Nutritional Status; PNI, Prognostic Nutritional Index; GNRI, Geriatric Nutritional Risk Index.

**Table 5 jcm-10-02247-t005:** Two different age-adjusted multivariate cox proportional risk regression analysis models to determine the predictors of one-year MACCEs, based on CONUT and PNI scores.

Variables	Model 2 Multivariate		Model 3 Multivariate	
	HR (95%CI)	*p*-Value	HR (95%CI)	*p*-Value
Age	1.005 (0.960–1.052)	0.82	1.027 (0.984–1.073)	0.22
BMI	1.007 (0.897–1.131)	0.91	1.004 (0.896–1.125)	0.94
Diabetes mellitus	1.852 (1.034–3.315)	0.04	2.072 (1.161–3.698)	0.01
eGFR	0.999 (0.983–1.015)	0.91	0.995 (0.979–1.011)	0.53
LVEF	0.919 (0.879–0.961)	<0.01	0.897 (0.861–0.934)	<0.01
CONUT score	1.434 (1.194–1.723)	<0.01	-	-
PNI score	-	-	0.979 (0.928–1.032)	0.43

Abbreviations: MACCEs, major adverse cardiac and cerebrovascular events; CONUT, The Controlling Nutritional Status; PNI, Prognostic Nutritional Index; BMI, body mass index; eGFR, estimated glomerular filtration rate; LVEF, left ventricular ejection fraction.

## Data Availability

The data presented in this study are available on request form the corresponding author. The data are not publicly available due to legal restrictions.

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
