# Peer review of "Predicting One-Year Deaths and Major Adverse Vascular Events with the Controlling Nutritional Status Score in Elderly Patients with Non–ST-Elevated Myocardial Infarction Undergoing Percutaneous Coronary Intervention"

_jcm, 2021, doi:10.3390/jcm10112247_

Round 1
Reviewer 1 Report
The novelty of this study is unclear, so the authors should clarify the novelty.
The authors showed the baseline characteristics of patients with ACS in Table 2, but not enough. Although the authors included the patients with diabetes, hypertension, dyslipidemia, and others in the current study, they have not shown the details of medication history. Therefore, the authors should provide their details. For example, antiplatelets, anticoagulants, calcium channel blockers, hypoglycemic agents, and lipid-lowering agents other than statins should be included in this study.
The authors should include BNP, NT-proBNP, hsCRP, inflammatory cytokines, and other inflammatory markers.
The authors should reanalysis after adding these data.
Author Response
Reviewer 1.
Q1. The novelty of this study is unclear, so the authors should clarify the novelty.
A1. Requested corrections have been made in line with your valuable suggestions and written in red.
Q2 and Q3. The authors showed the baseline characteristics of patients with ACS in Table 2, but not enough. Although the authors included the patients with diabetes, hypertension, dyslipidemia, and others in the current study, they have not shown the details of medication history. Therefore, the authors should provide their details. For example, antiplatelets, anticoagulants, calcium channel blockers, hypoglycemic agents, and lipid-lowering agents other than statins should be included in this study. The authors should include BNP, NT-proBNP, hsCRP, inflammatory cytokines, and other inflammatory markers.
A2. Requested corrections have been made in line with your valuable suggestions and written in red. However, since the tests suggested by you such as NT-proBNP, inflammatory cytokines, and other inflammatory markers were performed in very few of our patients, no additional adjustments could be made in terms of biochemical and hematological parameters.
Reviewer 2 Report
This is a retrospective study in which the authors compared three malnutrition scales in patients older than 60 years with non-ST-segment elevation infarction to predict the appearance of adverse events during 1 year of follow-up. Among the three tools analyzed, the CONUT scale, very simple to apply, is the most efficient. The clincal evolution of these patients could be improved by treating malnutrition. Comments:
- Authors should provide information on losses to follow-up
- The explanation about multivariate models is confusing to me. Age, as might be expected, is a factor associated with malnutrition and yet it seems that it does not remain in the multivariate models as an independent variable. This aspect needs to be clarified.
- The title would be more consistent with the content if it refers to ST-segment elevation infarcts
Author Response
Q1. Authors should provide information on losses to follow-up.
A1. Requested corrections have been made in line with your valuable suggestions and written in red.
Q2. The explanation about multivariate models is confusing to me. Age, as might be expected, is a factor associated with malnutrition and yet it seems that it does not remain in the multivariate models as an independent variable. This aspect needs to be clarified.
A2. Requested corrections have been made in line with your valuable suggestions and written in red.
In the previous analysis, age and GFR; They were not included in the multivariate analysis as they are components of the GRace score and would affect each other statistically negatively. But, age was added to the multivariate analysis in line with your suggestions. However, GRACE was excluded from the multivariate analysis as age correlates perfectly with the Grace score (r= 0.643, p<0.01) and GFR also correlates well with GRACE ( r= -0.488, p< 0.01), which would negatively affect each other's statistical significance. In addition, GRS is a validated and frequently used scoring system to predict 6-month mortality in NSTEMI patients, and its correlations with malnutrition-based scoring systems were evaluated and this was stated in the results section.
Q3. The title would be more consistent with the content if it refers to ST-segment elevation infarcts
A3. In line with your valuable suggestions, the title of the article has been corrected as " Predicting One-year Deaths and Major Adverse Vascular Events with the Controlling Nutritional Status Score in Elderly Patients with Non–ST-elevated Myocardial Infarction Under-going Percutaneous Coronary Intervention"..
Note: Professional support was received for English editing and the certificate has been uploaded to the system.

Round 2
Reviewer 1 Report
Although we have confirmed the revised manuscript, the authors need to correct minor points. There are still many miss typos in the revised manuscript, so the authors should check more carefully. They should confirm of half-width or full-width space of parentheses in all text, Figure legends, and tables. Also, italics or normal fonts and inappropriate positions of commas or periods are mixed in all text, so please check more carefully.
Author Response
Q1. Although we have confirmed the revised manuscript, the authors need to correct minor points. There are still many miss typos in the revised manuscript, so the authors should check more carefully. They should confirm of half-width or full-width space of parentheses in all text, Figure legends, and tables. Also, italics or normal fonts and inappropriate positions of commas or periods are mixed in all text, so please check more carefully.
A1. Requested corrections have been made in line with your valuable suggestions and written in red.
Note: Professional support was received for English editing and the certificate has been uploaded to the system.

Reviewer 2 Report
The authors adequately answer the questions posed, except for the question about study losses to follow-up.
Author Response
Q1. The authors adequately answer the questions posed, except for the question about study losses to follow-up.
A1. Requested corrections have been made in line with your valuable suggestions and written in red.
This manuscript is a resubmission of an earlier submission. The following is a list of the peer review reports and author responses from that submission.